# Topological minibands and interaction driven quantum anomalous Hall state in topological insulator based moiré heterostructures

Kaijie Yang[1], Zian Xu [2], Yanjie Feng[2], Frank Schindler[3], Yuanfeng Xu [4,5], Zhen Bi [1], B. Andrei Bernevig [5,6,7], Peizhe Tang [2,8] & Chao-Xing Liu [1,5] ✉

The presence of topological flat minibands in moiré materials provides an opportunity to explore the interplay between topology and correlation. In this work, we study moiré minibands in topological insulator films with two hybridized surface states under a moiré superlattice potential created by two-dimensional insulating materials. We show the lowest conduction (highest valence) Kramers' pair of minibands can be $\mathbb{Z}_2$ non-trivial when the minima (maxima) of moiré potential approximately form a hexagonal lattice with six-fold rotation symmetry. Coulomb interaction can drive the non-trivial Kramers' minibands into the quantum anomalous Hall state when they are half-filled, which is further stabilized by applying external gate voltages to break inversion. We propose the monolayer $Sb_2$ on top of $Sb_2Te_3$ films as a candidate based on first principles calculations. Our work demonstrates the topological insulator based moiré heterostructure as a potential platform for studying interacting topological phases.

Recent research interests have focused on the moiré superlattice in 2D Van der Waals heterostructures, including graphene[1–8] and transition metal dichalcogenide (TMD) multilayers[9–17], due to the strong correlation effect in the presence of flat bands. The flat bands formed by low-energy gapless Dirac fermions in magic angle twisted bilayer graphene typically have a bandwidth ~ 5 meV, much smaller than the band gap 25 ~ 35 meV that separates flat bands from higher energy bands and the Coulomb interaction of order 30 meV[2,3]. In contrast, the flat bands in TMD moiré heterostructures are formed by electrons with parabolic dispersion and have a typical bandwidth ~ 10 meV, separated by a comparable gap from other energy bands, and a huge on-site Coulomb interaction ~ 100 meV[10,11,18]. Besides the above materials, moiré superlattice has also been found in another family of van der

Waals heterostructures consisting of topological insulators (TIs)[19–28]. These TI-based moiré heterostructures show different features. TIs have the anomalous gapless surface bands that connect the bulk conduction and valence bands due to non-trivial bulk topology. The spin splitting of surface bands has a typical energy scale of hundreds meV due to the strong spin-orbit coupling (SOC). Previous studies[29–32] show that a single surface state remains gapless upon the moiré superlattice potential, leading to satellite Dirac cones and van Hove singularities, instead of isolated flat bands. Furthermore, the moiré superlattice in magnetic TI materials, e.g. $MnBi_2Te_4$, was predicted to host Chern insulator phase[33].

In this work, we studied a model of the TI thin film (e.g. $(Bi,Sb)_2Te_3$ film) with the moiré superlattice potential (See Fig. 1). Different from a

[1]Department of Physics, the Pennsylvania State University, University Park, PA 16802, USA. [2]School of Materials Science and Engineering, Beihang University, Beijing 100191, China. [3]Blackett Laboratory, Imperial College London, London SW7 2AZ, United Kingdom. [4]Center for Correlated Matter and School of Physics, Zhejiang University, Hangzhou 310058, China. [5]Department of Physics, Princeton University, Princeton, NJ 08544, USA. [6]Donostia International Physics Center, P. Manuel de Lardizabal 4, 20018 Donostia-San Sebastian, Spain. [7]IKERBASQUE, Basque Foundation for Science, Bilbao, Spain. [8]Max Planck Institute for the Structure and Dynamics of Matter and Center for Free Electron Laser Science, Hamburg 22761, Germany. ✉e-mail: cxl56@psu.edu

bulk TI, a strong hybridization between two surface states is expected for the TI thin film. The hybridization between two surface states can create isolated minibands that possess non-trivial $\mathbb{Z}_2$ topological invariant, denoted by $\nu$ below, in the low-energy moiré spectrum in a wide parameter space, particularly when the moiré potential approximately has six-fold rotation symmetry. In the presence of inversion symmetry, an emergent chiral symmetry in the low energy sector of surface states gives rise to $\nu_{CB1} + \nu_{V\ B1} = 1$ for the lowest Kramers' pair of conduction minibands, denoted as CB1, and the highest Kramers' pair of valence minibands, denoted as VB1, in Fig. 1d. We find $\nu_{CB1} = 1$, $\nu_{V\ B1} = 0$ ($\nu_{CB1} = 0$, $\nu_{V\ B1} = 1$) when the minima (maxima) of the moiré potential approximately form a hexagonal lattice. In the case of non-trivial CB1 ($\nu_{CB1} = 1$, $\nu_{V\ B1} = 0$), the lowest two Kramers' pairs of conduction minibands (CB1 and CB2 in Fig. 1d) together can be adiabatically connected to the Kane-Mele model[34] when increasing quadratic terms, and thus CB2 is also topologically non-trivial, $\nu_{CB2} = 1$. An asymmetric potential between two surface states can be generated by external gate voltages to break inversion but preserve six-fold rotation and generally induce the gap closing between different conduction minibands, leading to nodal phases. In the parameter regions where the conduction minibands are gapped from other minibands (parameter regions I, II, III in Fig. 2c), the CB1 is always topologically non-trivial, $\nu_{CB1} = 1$. We further study the influence of the Coulomb interaction via Hartree-Fock mean field theory when the CB1 carries $\nu_{CB1} = 1$ and is half filled, and find that the quantum anomalous Hall (QAH) state competes with a trivial insulator state in region I of Fig. 2c and it can be robustly energetically favored by the asymmetric potential in region II. Finally, we propose a possible experimental realization of the TI-based moiré heterostructure consisting of a monolayer $Sb_2$ layer on top of $Sb_2Te_3$ thin films based on the first principles calculations.

## Results

### Model Hamiltonian

We show a schematic of a heterostructure consisting of TI thin films and another 2D material (e.g. 2D Sb thin films) in Fig. 1a, b, and the moiré potential induced by the 2D material can affect both the top and bottom surface states with different strength. We assume the Fermi

energy is within the bulk gap of the TI thin film, and thus model this system with the Hamiltonian

$$
\begin{aligned}
H_0(\mathbf{r}) &= H^{TI} + H^M(\mathbf{r}), \\
H^{TI} &= v\tau_z(-i\partial_y s_x + i\partial_x s_y) + m\tau_x s_0, \\
H^M(\mathbf{r}) &= \frac{1+\alpha}{2}\Delta(\mathbf{r})\tau_0 s_0 + \frac{1-\alpha}{2}\Delta(\mathbf{r})\tau_z s_0 + V_0\tau_z s_0.
\end{aligned}
\tag{1}
$$

$H^{TI}$ denotes two surface states of a TI thin film with the inter-surface hybridization $m = m_0 + m_2(-\partial_x^2 - \partial_y^2)$, and $h_D^{t/b}(\mathbf{r}) = \pm v(-i\partial_y s_x + i\partial_x s_y)$ is the top/bottom surface Dirac Hamiltonian[35]. $s_{0,x,y,z}(\tau_{0,x,y,z})$ are the identity and Pauli matrices for spin (surfaces) and $v$ is the Fermi velocity. $H^M$ denotes the potential term, in which the $V_0$ term is the uniform asymmetric potential between two surfaces by gate voltages, the $\Delta(\mathbf{r})$ term is the moiré potential, and the $\alpha$ parameter ($0 \leq \alpha \leq 1$) represents the asymmetry between top and bottom surfaces. $\Delta(\mathbf{r})$ is real, spin-independent[29], and assumed to possess the $C_{3v}$ symmetry coinciding with the atomic crystal symmetry of TI thin films. With the basis of the Hamiltonian, the corresponding symmetry operators are $C_{3z} = \exp(-i\pi\tau_0 s_z/3)$ for three-fold rotation, $\mathcal{M}_y = \tau_0 s_y$ for y-directional mirror, and $\mathcal{T} = i\tau_0 s_y\mathcal{K}$ with $\mathcal{K}$ as complex conjugate for time-reversal (TR). The moiré superlattice potential can be expanded as

$$
\Delta(\mathbf{r}) = \sum_{\mathbf{G}} \Delta_{\mathbf{G}} e^{i\mathbf{G}\cdot\mathbf{r}},
\tag{2}
$$

where $\mathbf{G} = n_1\mathbf{b}_1^M + n_2\mathbf{b}_2^M$ is the moiré reciprocal lattice vectors with $\mathbf{b}_1^M = \frac{4\pi}{\sqrt{3}|\mathbf{a}_1^M|}(1/2, \sqrt{3}/2)$, $\mathbf{b}_2^M = \frac{4\pi}{\sqrt{3}|\mathbf{a}_1^M|}(-1/2, \sqrt{3}/2)$ and $n_{1,2}$ as integers. $\mathbf{a}_{1,2}^M$ are the primitive vectors for moiré superlattice (see Fig. 1c). The uniform part $\Delta_{\mathbf{G}=\mathbf{0}}$ can be absorbed into the chemical potential $\mu$ and the asymmetric potential $V_0$. To the lowest order, we only keep the first shell reciprocal lattice vectors $\pm\mathbf{b}_1^M$, $\pm\mathbf{b}_2^M$, $\pm(\mathbf{b}_1^M - \mathbf{b}_2^M)$, as shown in Fig. 1d. The values of $\Delta_{\mathbf{G}}$ for different $\mathbf{G}$s are connected by three-fold rotation $C_{3z}$ and $\mathcal{T}$, so there is only one independent complex

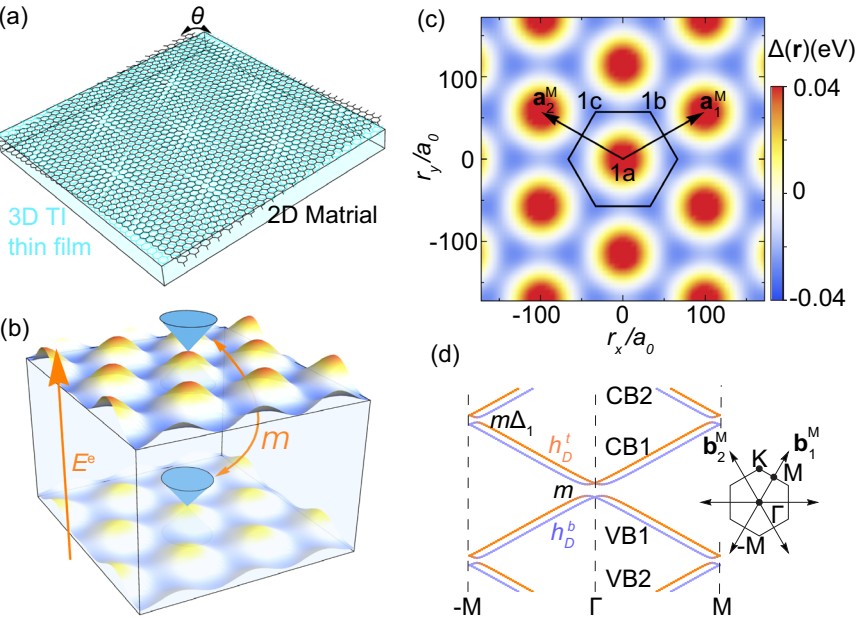

**Fig. 1 | TI-based moiré heterostructures and moiré minibands. a** A schematic figure for the twisted 2D materials (black) on top of a topological insulator thin film (cyan) with an angle $\theta$. **b** Schematic illustration of the moiré potentials from twisted 2D materials on the top and bottom surface of a TI thin film. The blue Dirac cones represent the top and bottom surface states coupled by $m$. An out-of-plane external electrical field $E^e$ creates the potential $V_0$. **c** The moiré potential $\Delta(\mathbf{r})$ with $\phi = 0$. $\mathbf{a}_1^M$, $\mathbf{a}_2^M$ are primitive vectors for a moiré unit cell. $1a$, $1b$, $1c$ are Wyckoff positions under the point group $C_{3v}$. **d** Schematic view of the spectrum. The orange (blue) lines are top (bottom) surface Dirac cones at $\Gamma$. Inset is the moiré BZ with the first shell moiré reciprocal lattice vectors.

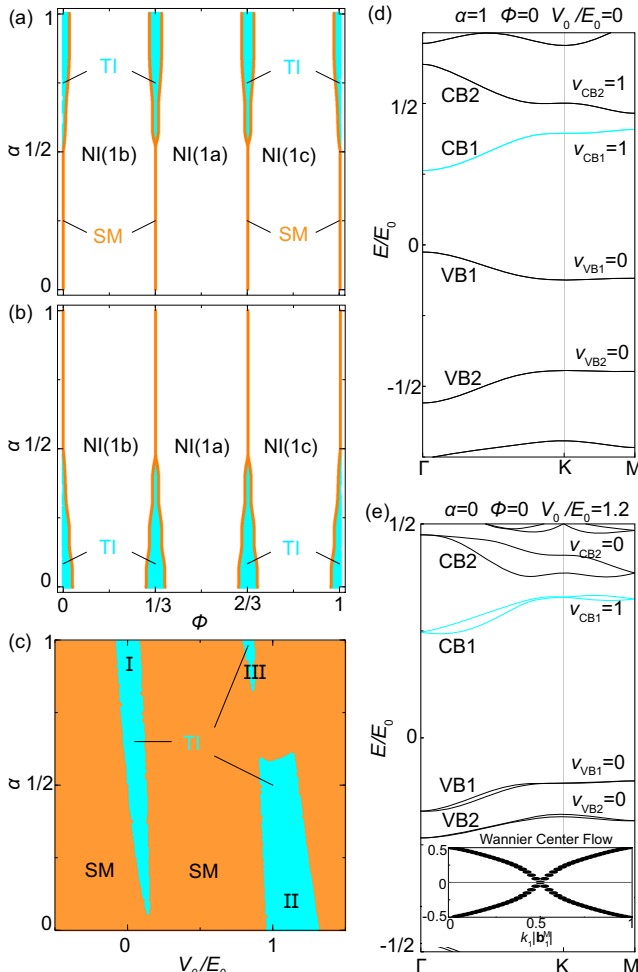

**Fig. 2 | Topological moiré minibands and phase diagram. a, b** The topological phase diagrams of the lowest conduction bands CB1 for different moiré potentials with $V_0/E_0 = 0$ for (**a**) and $V_0/E_0 = 1.2$ for (**b**). The three phases for CB1 are topological insulator (TI) phase with $\nu_{CB1} = 1$, normal insulator (NI) phase with $\nu_{CB1} = 0$ and semi-metal (SM) phase with CB1 connected to higher energy bands. The 1a, 1b, 1c are the Wyckoff positions of corresponding atomic orbitals for CB1 in NI phases. **c** The phase diagram for different uniform asymmetrical potentials with $\phi = 0$. Regions I, II and III are three parameter regimes with $\nu_{CB1} = 1$ for CB1. **d, e** Examples of spectra with nontrivial CB1 in the regions I and II, respectively. The spectrum in (**d**) has both TR and inversion, and is thus doubly degenerate. The inset of (**e**) is the Wannier center flow for CB1.

parameter, chosen to be $\Delta_{\mathbf{b}_1^M} = \Delta_1 e^{i2\pi\phi}$, where $\Delta_1$ is real and $\phi$ is the phase that tunes the relative strengths of potentials at three Wyckoff positions $1a, 1b, 1c$ in one moiré unit cell. Figure 1c shows the moiré potential at $\phi = 0$ with an additional six-fold rotation symmetry $C_{6z} = \exp(-i\pi\tau_0 s_z/6)$, and the corresponding potential minima form the multiplicity-2 Wyckoff positions of the hexagonal lattice. The parameters used in our calculations below are $|\mathbf{a}_1^M| = 28$ nm, $E_0 = \upsilon|\mathbf{b}_1^M| = 38.5$ meV[36], $m_0 = 0.4E_0$, $\Delta_1 = 0.24E_0$. The $m_2$ term and other quadratic terms are negligible for the low energy minibands in realistic materials as the relevant energy scale is around 1 meV with a typical moiré momentum $10^{-2}$ Å$^{-1}$, much smaller than other terms in $H^{TI}$. But we still keep this term in low energy Hamiltonian as it plays an important role for connecting this model to the Kane-Mele model discussed below.

## $\mathbb{Z}_2$ nontrivial moiré minibands

We first illustrate the crucial role of inter-surface hybridization in inducing isolated moiré minibands in TI thin films through the schematic view of the spectrum in Fig. 1d. For a single Dirac surface state, it is known[29,31,37] that moiré potential can fold the Dirac dispersion and the band touchings at the TR-invariant momenta, e.g. Γ and $M$, in the moiré Brillouin zone (BZ) remain gapless due to the Kramers' theorem of TR symmetry. This leads to satellite Dirac cones, but prevents the formation of gaps and hence isolated moiré minibands. For TI thin films, the inter-surface hybridization $m$ can directly produce a gap at Γ while its combined effect with the moiré potential $\Delta(\mathbf{r})$ can lead to a gap (proportional to $m\Delta_1$) at $M$ (Fig. 1d). The gap openings at both Γ and $M$ lead to the isolated moiré minibands, as demonstrated in Fig. 2d, e for the moiré spectrum of the model Hamiltonian Eq. (1) with different sets of parameters. The bandwidth of isolated bands can be significantly reduced by increasing $m$, $\Delta_1$ and the length of moiré unit cells. (See Supplementary Note 1A)

We are interested in the possibility of realizing $\mathbb{Z}_2$-nontrivial moiré minibands, particularly the low-energy Kramers' pairs of conduction (valence) minibands, labelled by CB1, CB2 (VB1, VB2) in Fig. 2d, e. For the parameters in Fig. 2d, CB1 and CB2 are topologically non-trivial while VB1 and VB2 are trivial ($\nu_{CB1} = \nu_{CB2} = 1$, $\nu_{VB1} = \nu_{VB2} = 0$). For the parameters in Fig. 2e, only CB1 is non-trivial while other minibands are trivial ($\nu_{CB1} = 1$, $\nu_{CB2} = \nu_{VB1} = \nu_{VB2} = 0$). Figure 2a, b show the $\mathbb{Z}_2$-invariant $\nu_{CB1}$ for CB1 as a function of $\alpha$ and $\phi$ for a fixed $V_0/E_0 = 0$ and 1.2, respectively. The blue regions correspond to $\nu_{CB1} = 1$ while the white regions to $\nu_{CB1} = 0$, and these two regions are separated by metallic lines (orange color). For both $V_0$ values, the $\nu_{CB1} = 1$ blue regions appear around $\phi = 0, 1/3, 2/3$. At these $\phi$ values, there is an additional $C_{6z}$ rotation symmetry, leading to a hexagonal lattice with the $C_{6\upsilon}$ group. Moreover, $\phi$ and $\phi + 1/3$ are equivalent up to a translation with the vector $\mathbf{a}_1^M/3 + 2\mathbf{a}_2^M/3$ (See Supplementary Note 1B). Thus, we only show $\nu_{CB1}$ as a function of $\alpha$ and $V_0$ for $\phi = 0$ in Fig. 2c and find three different parameter regions I, II, III with $\nu_{CB1} = 1$. These topologically non-trivial regions are separated by semi-metal phases that have band touchings between CB1 and CB2. $\nu_{CB1}$ for other $\phi$ is discussed in Supplementary Note 1B and normal insulator phases are discussed in Supplementary Note 1D.

The region I can be adiabatically connected to the parameter set $\alpha = 1$, $V_0/E_0 = 0$ with the band dispersion shown in Fig. 2d, where the inversion symmetry $\mathcal{I} = \tau_x s_0$ and the horizontal mirror symmetry $\mathcal{M}_z = -i\tau_x s_z$ are present ($D_{6h}$ group). The $C_{6z}$ symmetry leads to the existence of the inversion symmetry by $\mathcal{I} = C_{6z}^3 \mathcal{M}_z$. From the Fu-Kane parity criterion[38], the $\mathbb{Z}_2$-invariant $\nu$ can be determined by $(-1)^\nu = \prod_i \lambda_{\Gamma_i}$ and $\lambda_{\Gamma_i}$ is the parity of eigenstates at the TR invariant momenta $\Gamma_{i=1,...,4}$, which correspond to one Γ point and three $M$ points in 2D moiré BZ. The values of parities $\lambda_{\Gamma_i}$ can be derived analytically in the weak $\Delta_1$ limit and depend on the hybridization $m$ and the moiré potential strength $\Delta_1$ (See Supplementary Note 1A). From the analytical solutions, we find CB1 and VB1 have the same parity at $M$ ($\lambda_M^{CB1} = \lambda_M^{VB1}$) but opposite parities at Γ ($\lambda_\Gamma^{VB1} = -\lambda_\Gamma^{CB1}$), resulting in $\nu_{CB1} + \nu_{VB1} = 1$ mod 2, implying that one of them is $\mathbb{Z}_2$-nontrivial while the other is trivial. As discussed in Supplementary Note 1A, the relation of $\mathbb{Z}_2$ invariant between the CB1 and VB1 minibands can be understood as the consequence of the emergent chiral symmetry operator $\mathcal{C} = \tau_z s_z$ of $H^{TI}$.

At $\phi = 0$ and $\alpha = 1$ in Fig. 2d, we notice that the CB2 minibands are also topologically non-trivial ($\nu_{CB2} = 1$), so $\nu_{CB1} + \nu_{CB2} = 0$ mod 2. According to the irreducible representations of CB1 and CB2 at high-symmetry momenta (See Supplementary Note 1C), these two minibands can together form an elementary band representation (EBR) $\bar{E}_1^{2b} \uparrow G$ induced in the space group $P6mm$[39,40], which corresponds to the atomic limit with two s-wave atomic orbitals at the symmetry-related Wyckoff positions $1b$ and $1c$ in Fig. 1(c). Once CB1 is isolated from CB2, CB1 itself does not have an atomic limit preserving $C_{6z}$

symmetry and is thus topological. If $\mathcal{C}_{6z}$ symmetry is relaxed, CB1 and CB2 instead correspond to the reducible band representation $^1\bar{E}^2\bar{E} + \bar{E}_1 \uparrow G'$ with $G'$ to be $P3m1$, and correspondingly, they can be gapped. Indeed, as demonstrated in Supplementary Note 1C, when the $m_2$ term is tuned to dominate over other terms in $H_0$, we can adiabatically connect the CB1 and CB2 together in Fig. 2d to the effective Kane-Mele model[34]. This provides an alternative explanation of nontrivial $\mathbb{Z}_2$ numbers for both CB1 and CB2 in Fig. 2d.

For the nontrivial region II in Fig. 2c, we consider the parameter set $\phi = 0$, $\alpha = 0$, $V_0/E_0 = 1.2$ with the energy dispersion shown in Fig. 2e. The Fu-Kane criterion cannot be applied as inversion is broken, so we directly calculate the Wannier center flow[41] for the CB1 in the inset of Fig. 2e, which corresponds to $\nu_{CB1} = 1$. Different from the case of Fig. 2d, CB2 is now topologically trivial $\nu_{CB2} = 0$. We also examine the band evolution with respect to $m_2$ in the model, which is quite different from the case with inversion symmetry, as discussed in Supplementary Note 1C. When the $m_2$ term dominates in $H_0$, CB1 and CB2 can be mapped to the Kane-Mele model with a Rashba SOC term from the inversion symmetry breaking, which leads to the gap closing between CB1 and CB2 around $K$ in moiré BZ. The overall $\mathbb{Z}_2$ number $\nu_{CB1} + \nu_{CB2}$ mod 2 is 0 because CB1 and CB2 together also form a EBR coming from s-wave atomic orbitals located at the two potential minima of the $\mathcal{C}_{6z}$ symmetric moiré potential. When reducing $m_2$, a Dirac type of gap closing between CB2 and higher-energy conduction minibands occurs at certain critical value of $m_2$ and changes $\nu_{CB1} + \nu_{CB2}$ mod 2 to 1, which is persisted to $m_2 = 0$ ($\nu_{CB1} = 1$ and $\nu_{CB2} = 0$). The other $\mathbb{Z}_2$ non-trivial minibands are found to appear in a much higher energy when $m_2$ is small (See Supplementary Fig. 6 in Supplementary Note 1C). This is in sharp contrast to the inversion-symmetric case in which CB1 and CB2 together have $\nu_{CB1} + \nu_{CB2} = 0$ mod 2 when varying $m_2$.

## Interaction-driven QAH state

The Coulomb interaction of electrons in the moiré superlattice can be estimated as $U_0 = e^2/4\pi\varepsilon_0\varepsilon_r|\mathbf{a}_1^M| \approx 5.11$ meV $\sim 0.13E_0$, in which $e$ is the electron charge, $\varepsilon_0$ is vacuum permittivity, and dielectric constant $\varepsilon_r$ is about 10.[42] The value of $U_0$ is comparable to both the moiré miniband width $\sim 0.1E_0 \approx 3.85$ meV and miniband gaps $\sim 0.1E_0$. We next study the effects of the Coulomb interaction with the Hartree-Fock mean-field theory[42–47]. We first project the moiré Hamiltonian and the Coulomb interaction into the low-energy subspace spanned by either CB1 (a two-band model) or both CB1 and CB2 (a four-band model). By treating the density matrix $\rho_{n_1 n_2}(\mathbf{k}) = \langle c_{n_1}^\dagger(\mathbf{k})c_{n_2}(\mathbf{k})\rangle$ as the order parameter with $c_n^\dagger(\mathbf{k})$ for the creation operator of the $n$th eigenstate in the two-band or four-band subspace, we can decompose the Coulomb interaction Hamiltonian into two-fermion terms so that the order parameter $\rho(\mathbf{k})$ can be solved self-consistently (See Supplementary Note 2).

In the two-band model, we generally consider two types of order parameters, (1) $\rho_z(\mathbf{k}) \propto f_z(\mathbf{k})\sigma_z$ and (2) $\rho_{xy}(\mathbf{k}) \propto f_x(\mathbf{k})\sigma_x + f_y(\mathbf{k})\sigma_y$, where the $\sigma$ matrix is for the Kramers' pair of CB1 and $f_{x,y,z}(\mathbf{k})$ represents the momentum-dependent part of the order parameter. The order parameter $\rho_0 \propto \sigma_0$ is directly related to the band occupation and we always consider half-filling for the Kramers' pair bands of CB1. At $\phi = 0$, $\alpha = 1$, $V_0/E_0 = 0$, the horizontal mirror symmetry $\mathcal{M}_z$ is present and the non-interacting Hamiltonian possesses $D_{6h}$ group symmetry, so two spin states of CB1 can be labelled by the mirror eigen-values $\pm i$, and the $\sigma$ matrices of the order parameter $\rho$ is written under the $\mathcal{M}_z$ eigenstates. The two mirror-eigenstates carry nonzero mirror Chern number $\pm 1$ from the nontrivial $\mathbb{Z}_2$ topology. Thus, $\rho_z(\mathbf{k})$ and $\rho_{xy}(\mathbf{k})$ correspond to the mirror-polarized and mirror-coherent ground states.

The mirror polarized state ($\rho_z(\mathbf{k})$) spontaneously breaks $\mathcal{C}_{2z}\mathcal{T}$ symmetry relating two mirror eigenstates, while the mirror coherent state ($\rho_{xy}(\mathbf{k})$) breaks $\mathcal{M}_z$ by superposition of two mirror eigenstates with a fixed relative U(1) phase between them (See Supplementary

Note 2C). The self-consistent calculations suggest that both $\rho_z(\mathbf{k})$ and $\rho_{xy}(\mathbf{k})$ can be non-zero solutions when the Coulomb interaction exceeds certain critical values $U_c \sim 0.05E_0 \approx 1.92$ meV, as shown in Fig. 3c, where the ground state energies of self-consistent $\rho_z(\mathbf{k})$ and $\rho_{xy}(\mathbf{k})$ are shown as a function of interaction strength $U(\mathbf{a}_1^M)$, which is treated as a tuning parameter and equal to $U_0$ for the realistic moiré superlattice. Our estimate of Coulomb interaction $0.13E_0$ in TI moiré systems is larger than this critical value. From Fig. 3c, we also see that the mirror-polarized state $\rho_z(\mathbf{k})$ has a lower ground state energy than the mirror-coherent state $\rho_{xy}(\mathbf{k})$. The energy spectrum of the CB1 before (blue lines) and after (orange lines) taking into account the $\rho_z(\mathbf{k})$ order parameter is shown in Fig. 3a, in which the metallic state of CB1 (blue lines) is fully gapped out by $\rho_z(\mathbf{k})$ at half-filling. With the order parameter $\rho_z(\mathbf{k})$ (orange lines in Fig. 3a), the ground state only fills the lower energy band while the higher band is the excited spectrum within the Hartree-Fock approximation. Due to non-zero mirror Chern number of non-interacting CB1 states, the mirror-polarized state $\rho_z(\mathbf{k})$ carries Chern number $+1$ of the lower band and thus gives rise to the QAH state. As shown in Fig. 3b (also Supplementary Note 2C), the mirror coherent state $\rho_{xy}(\mathbf{k})$ opens gaps at TR invariant momenta $\Gamma$ and M by spontaneously breaking $\mathcal{T}$ but has nodes at $K$ due to the $C_{2z}\mathcal{T}$ symmetry. This explains why the mirror-polarized state has a lower ground state energy than the mirror-coherent state. Besides all the

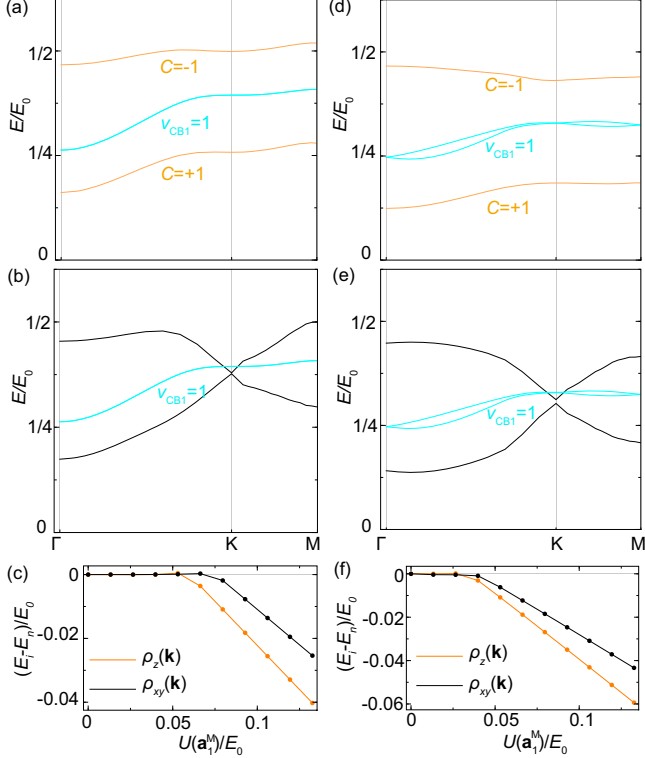

**Fig. 3 | Hartree-Fock ground energies and ground states. a** The spectra (orange) for the Hartree-Fock mean-field Hamiltonian with the order parameter $\rho_z(\mathbf{k})$ at half filling of CB1 for the case with $\phi = 0$, $\alpha = 1$, $V_0/E_0 = 0$. $C$ is the Chern number of each band. **b** The Hartree-Fock spectra (black) for the order parameter $\rho_{xy}(\mathbf{k})$ for the same parameter as (**a**). In (**a**, **b**), the blue lines are single-particle spectra. **c** The difference in energy per particle between the self-consistent Hartree-Fock states $E_i$ and the non-interacting state $E_n$ as a function of Coulomb interaction strengths for the order parameters $\rho_z(\mathbf{k})$ (orange) and $\rho_{xy}(\mathbf{k})$ (black). **d** The spectra (orange) for the Hartree-Fock mean-field Hamiltonian with the order parameter $\rho_z(\mathbf{k})$ at half filling of CB1 for the case with $\phi = 0$, $\alpha = 0$, $V_0/E_0 = 1.2$. **e** The Hartree-Fock spectra (black) for the order parameter $\rho_{xy}(\mathbf{k})$ for the same parameter as (**d**). In (**d**, **e**), the blue lines are single-particle spectra. **f** The energy difference $E_i - E_n$ for the order parameters $\rho_z(\mathbf{k})$ (orange) and $\rho_{xy}(\mathbf{k})$ (black).

uniform order parameters, a nonuniform magnetic order parameter[32] is also examined in Supplementary Note 2F, and is found to possess a larger critical interaction strength compared to the QAH phase at the half filling of CB1. Thus, the mirror-polarized QAH state can be driven by Coulomb interaction in this system.

We also study the case of $\phi = 0$, $\alpha = 0$, $V_0/E_0 = 1.2$ within the two-band model, in which the mirror $\mathcal{M}_z$ is broken at the single-particle level and six-fold rotation remains, in Supplementary Note 2C and find the $\rho_z(\mathbf{k})$ is still energetically favored, as shown in Fig. 3f. The spectra with the order parameter $\rho_z(\mathbf{k})$, $\rho_{xy}(\mathbf{k})$ are shown in Fig. 3d, e, respectively. The ground state is a Chern insulator.

As the miniband gap is comparable to Coulomb interaction, one may ask if the inter-miniband mixing due to Coulomb interaction can change the topological nature of the ground state. Thus, we study the Coulomb interaction effect in a four-band model including both CB1 and CB2, as discussed in Supplementary Note 2D. For the inversion-symmetric case $\phi = 0$, $\alpha = 1$, $V_0/E_0 = 0$, the ground state of the four-band model is still the mirror polarized $C = \pm 1$ state in regime B (blue) of Fig. 4a, when $U(\mathbf{a}_1^M) = 0.08E_0$ is smaller than the miniband gap - $0.1E_0$, with the spectra shown in Fig. 4c. When $U(\mathbf{a}_1^M) = 0.13E_0$ is larger than the miniband gap (regime C (brown) of Fig. 4a), the strong Coulomb interaction can induce mixing between CB1 and CB2 within one mirror parity sector and drive a topological phase transition to the $C = 0$ state shown in Fig. 4d (More details in Supplementary Note 2D). However, the situation for the inversion-asymmetric case $\phi = 0$, $\alpha = 0$, $V_0/E_0 = 1.2$ is different as $\nu_{CB1} = 1$ and $\nu_{CB2} = 0$. For the realistic estimated value $U(\mathbf{a}_1^M) \approx 0.13E_0$ that is larger than miniband gap, the interacting ground state of the four-band model carries $C = \pm 1$ and thus remains the same as that of the two-band model, as shown by the regime B (blue) in Fig. 4b. The energy spectra in this case is shown in Fig. 4e. By comparing the phase diagrams for the inversion symmetric and asymmetric cases, we conclude that the asymmetric potential $V_0$ stabilizes the interaction-driven QAH state in TI moiré heterostructures.

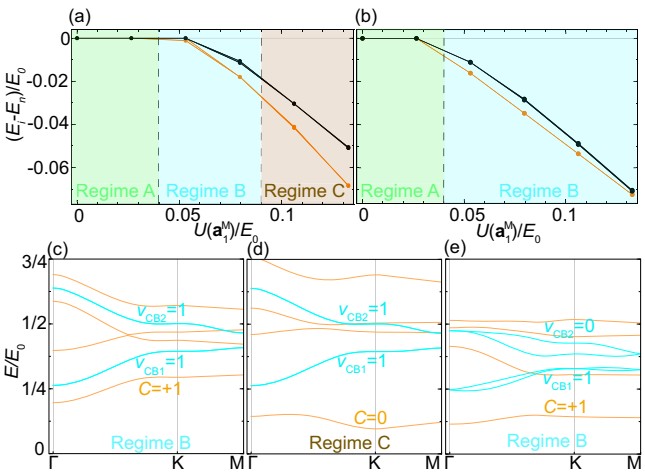

**Fig. 4 | Hartree-Fock interacting phase diagrams. a** The energy difference per particle $E_i - E_n$ at 1/4 filling of the four-band model with both CB1 and CB2 for the case $\phi = 0$, $\alpha = 1$, $V_0/E_0 = 0$. Here $E_i$ and $E_n$ is the interacting ground state energy and non-interacting metallic state energy, respectively. The orange (black) line is for the $C_2\mathcal{T}$ symmetry breaking (preserving) density matrix. The interacting ground states in the regime A, B, C correspond to a metallic phase, an insulating phase with $C = \pm 1$, and an insulating phase with $C = 0$, respectively. **b** $E_i - E_n$ for the case with $\phi = 0$, $\alpha = 0$, $V_0/E_0 = 1.2$. **c, d** The spectra (orange) of the Hartree-Fock mean-field Hamiltonian for the Coulomb interaction strength in regime B and C of (**a**). $C$ is the Chern number of each band. **e** The spectra (orange) of the mean-field Hamiltonian for the Coulomb interaction strength in regime B of (**b**). In (**c–e**), the blues lines are single-particle spectra.

## Sb$_2$/Sb$_2$Te$_3$ moiré heterostructure

We propose a possible experimental realization of a TI based moiré heterostructure with twisted Sb$_2$ monolayer on top of Sb$_2$Te$_3$ thin film. The moiré lattice structure is shown in Fig. 5a. Sb$_2$Te$_3$ is a prototype of three dimensional TI with layered structures. Within one quintuple layer (QL, see the red and blue dots in Fig. 5a), there is strong chemical binding formed by the sequential Te-Sb-Te-Sb-Te atomic layers and the van der Waals coupling is between adjacent QLs[48]. Precise control of layer thickness of the Sb$_2$Te$_3$ thin film has been achieved via molecular beam epitaxy (MBE) method experimentally[49,50]. On the top of Sb$_2$Te$_3$ thin film, Sb$_2$ monolayer could be deposited[19,51,52], forming a Sb$_2$/Sb$_2$Te$_3$ heterostructure. By using density functional theory (DFT) calculations, we confirm that Sb$_2$ monolayer with buckled honeycomb structure marked as the gray in Fig. 5a is a semiconductor with a band gap larger than that of Sb$_2$Te$_3$ thin films. Furthermore, we put Sb$_2$ monolayer on the top of 2QL Sb$_2$Te$_3$ thin films with different stackings, including the AA, AB, and BA stackings (see Fig. 5a). The corresponding electronic band structures are shown in Fig. 5c. The work function of monolayer Sb$_2$ and Sb$_2$Te$_3$ thin film matches with each other, forming the type I semiconductor hetero-junction. Around the Fermi level, the conduction and valence bands are both mainly contributed by two strongly hybridized surface states of the 2QL Sb$_2$Te$_3$ thin film. The role of Sb$_2$ monolayer is to provide a potential along the out-of-plane direction, leading to a Rashba type of spin-split bands. Thus, the twisted Sb$_2$/Sb$_2$Te$_3$ moiré heterostructure satisfies the requirements mentioned above for the $\mathbb{Z}_2$ nontrivial moiré minibands.

To connect the theoretical moiré model Hamiltonian in Eq. (1) to electronic band structure from DFT calculations, we first introduce a uniform shifting vector $\mathbf{d_R}$ between monolayer Sb$_2$ and 2QL Sb$_2$Te$_3$ thin film, and AA, AB, and BA stackings correspond to $\mathbf{d_R} = 0$, $\tilde{\mathbf{a}}_1/3 + 2\tilde{\mathbf{a}}_2/3$, and $2\tilde{\mathbf{a}}_1/3 + \tilde{\mathbf{a}}_2/3$, respectively (Fig. 5b). $\tilde{\mathbf{a}}_{1,2}$ are atomic primitive lattice vectors for the Sb$_2$Te$_3$ lattice shown in Fig. 5b. The spectrum from DFT calculations with different stackings is fitted by the dispersion of two-surface-state atomic Hamiltonian

$$H^{DFT}(\mathbf{k},\mathbf{d_R}) = H^{TI}(\mathbf{k}) + \frac{1+\alpha}{2}\tilde{\Delta}(\mathbf{d_R})\tau_0 s_0 + \frac{1-\alpha}{2}\tilde{\Delta}(\mathbf{d_R})\tau_z s_0, \tag{3}$$

where $s_0(\tau_{0,z})$ are the Pauli matrices for the spin (surfaces). $\tilde{\Delta}(\mathbf{d_R})$ is a uniform atomic potential induced by the Sb$_2$ monolayer for a fixed $\mathbf{d_R}$ and different $\mathbf{d_R}$ values correspond to different stacking configurations, shown in Fig. 5b. For the $\mathbf{d_R}$ values corresponding to the AA, AB, BA and several other stackings in Supplementary Note 3, we fit the energy dispersion of the model Hamiltonian $H^{DFT}(\mathbf{k}, \mathbf{d_R})$ to that from the DFT calculations as shown in orange lines in Fig. 5c, which fits well with the conduction bands of the surface states. From fitting, we can extract $\tilde{\Delta}(\mathbf{d_R})$, which can be further interpolated as a continuous function of $\mathbf{d_R}$ shown in Fig. 5d. $\tilde{\Delta}(\mathbf{d_R})$ has the periodicity of the atomic unit-cell defined by $\tilde{\mathbf{a}}_{1,2}$. All other parameters in $H^{DFT}(\mathbf{k}, \mathbf{d_R})$ are treated as constants and can also obtained by fitting to the DFT bands. After obtaining the parameters for $H^{DFT}(\mathbf{k}, \mathbf{d_R})$, the next step is to connect them to those of the moiré Hamiltonian $H_0$ in Eq. (1). For the moiré TI with the twist angle $\theta$, the local shift between two layers at the atomic lattice vector $\mathbf{R}$ of the Sb$_2$Te$_3$ layer is $\mathbf{d_R} = \mathcal{R}(\theta)\mathbf{R} - \mathbf{R}$, where $\mathcal{R}(\theta)$ is the rotation operator, so we can obtain the potential

$$\Delta(\mathbf{R}) \approx \tilde{\Delta}(\mathbf{d_R}) \tag{4}$$

at the location $\mathbf{R}$. The last step is to treat $\Delta(\mathbf{r})$ as a function of continuous $\mathbf{r}$ by interpolating the function $\Delta(\mathbf{R})$ (See Supplementary Note 4), and $\Delta(\mathbf{r})$ serves as the morié superlattice potential for the model Hamiltonian $H_0(\mathbf{r})$. Besides, all the other parameters in $H_0$ are chosen to be the same as those in $H^{DFT}$. In Fig. 5d, the potential maximum of $\tilde{\Delta}(\mathbf{d_R})$ appears at the AB stacking while two local minima exist

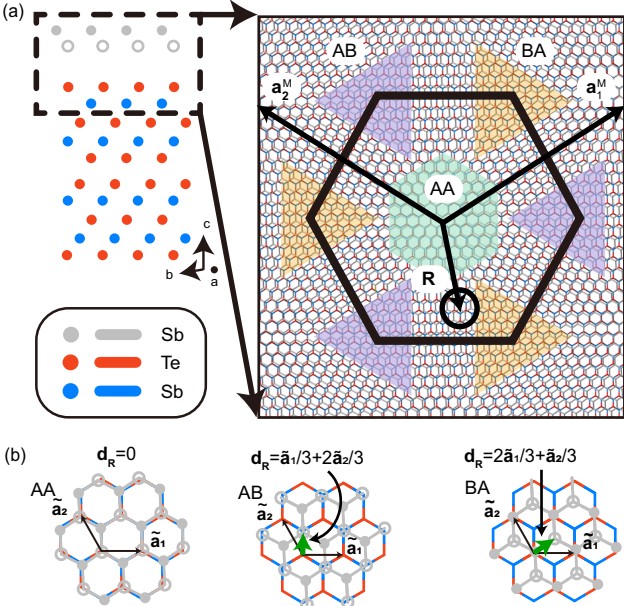

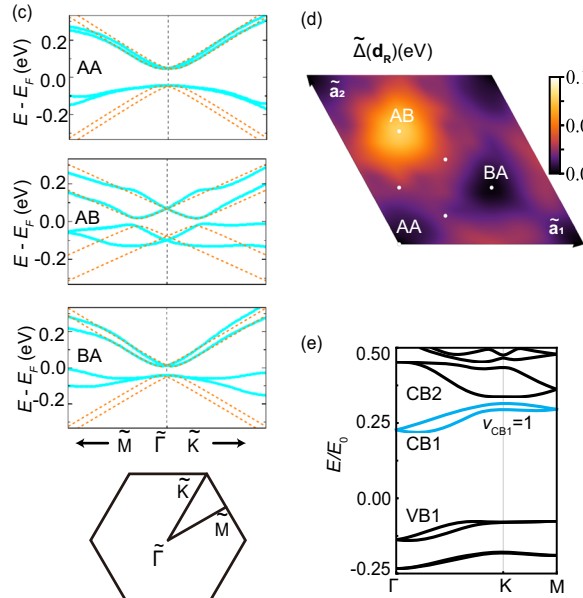

**Fig. 5 | DFT calculations of Sb$_2$ on top of 2QL Sb$_2$Te$_3$ films. a** Side view of the Sb$_2$/2QL Sb$_2$Te$_3$ heterostructure with AA stacking (left panel) and the moiré pattern for twisted Sb$_2$ on top of Sb$_2$Te$_3$ thin film (right panel). To show the moiré pattern clearly, we only plot atoms in the region marked by black dashed lines in the left panel. The regions with green, purple, and yellow background label structures with AA, AB, and BA stacking respectively. The primitive vectors for moiré supercell $\mathbf{a}_1^M$ and $\mathbf{a}_2^M$ are marked by black arrows. **b** Top views of configurations with AA, AB, BA stacking, respectively. The atomic primitive lattice vectors of the 2QL Sb$_2$Te$_3$ thin film are labeled as $\tilde{\mathbf{a}}_1$ and $\tilde{\mathbf{a}}_2$. The green arrow labels the shift $\mathbf{d_R}$ between the Sb$_2$Te$_3$ layer and Sb$_2$ monolayer in each stacked configuration. **c** Band structures (blue lines) around the $\bar{\Gamma}$ point for heterostructures with AA, AB, and BA stacking from DFT calculations. The orange lines are fitted spectra from the two-surface-state atomic Hamiltonian in Eq. (3). The Brillouin zone is plotted for the slab model used in DFT calculations with atomic primitive lattices. The Fermi levels are set as zero. **d** The superlattice potential $\tilde{\Delta}(\mathbf{d_R})$ as a function of $\mathbf{d_R}$ shown in the moiré superlattice. $\tilde{\mathbf{a}}_{1,2}$ are marked by the black arrows. **e** Energy spectrum for twisted monolayer Sb$_2$ and 2QL Sb$_2$Te$_3$ with the superlattice potential shown in (**d**).

at the BA and AA stackings and are close in energy. The parameters for the moiré potential at $\theta = 0.5°$ is given by $\Delta_1/E_0 = 0.22$, $\alpha = 0.16$, and $\phi = 0.68\pi$, close to $\phi = 2\pi/3$ for the $C_{6z}$-rotation symmetric potential. Figure 5e shows the energy dispersion of moiré minibands for $V_0/E_0 = 1.2$, in which the lowest conduction bands (cyan) indeed are isolated minibands with nontrivial $\nu_{CB1}=1$.

## Discussion

In summary, we demonstrate that the superlattice potential in a TI thin film can give rise to $\mathbb{Z}_2$ non-trivial isolated moiré minibands and Coulomb interaction can drive the system into the QAH state when the Kramer's pair of non-trivial minibands are half filled. Besides the twisted Sb$_2$ monolayer on top of the Sb$_2$Te$_3$ thin film, our model can be generally applied to other TI heterostructures with the in-plane superlattice potential, which can come from either the moiré pattern of another 2D insulating material or gating a periodic patterned dielectric substrate[53–57]. The 2D TI thin films can be in a quantum spin Hall state or trivial insulator state, depending on the relative sign between $m_0$ and $m_2$ in the model Hamiltonian (see Eq. (1))[58]. Our calculations suggest that the moiré potential can lead to $\mathbb{Z}_2$ non-trivial minibands no matter the sign of $m_2$, once this term is negligible compared to the linear term in the moiré scale. Such a result implies the possibility of realizing isolated $\mathbb{Z}_2$ non-trivial minibands in other 2D topologically trivial systems with strong Rashba SOC. In our calculation, a large moiré superlattice constant ($|\mathbf{a}_1^M| \sim 28$ nm) leads to small energy scales, around a few meV, for miniband widths, miniband gaps and Coulomb interactions, which may be disturbed by disorders. The miniband topological property is robust against disorder when the disorder strength is smaller than the miniband width (~2.2 meV). In Supplementary Note 2E, we reduce $|\mathbf{a}_1^M|$ to ~14 nm, which yields larger energy scales (around 10 meV) of minibands and Coulomb interaction, and our Hartree-Fock calculations suggest the estimated Coulomb

interaction is still strong enough to drive the system into the QAH state. For a smaller moiré lattice constant $|\mathbf{a}_1^M|$, it is desirable to reduce the bandwidth of moiré minibands while keeping the Coulomb energy, and this can be achieved by twisting two identical TIs or with in-plane magnetization, as proposed recently[30,59]. Moreover, at larger moiré superlattice constant, the lattice reconstruction may occur in the moiré superlattice (See Supplementary Note 3).

## Data availability

The data for the non-interaction spectra, Hartree-Fock mean-field spectra, interacting phase diagram is available in Zenodo at https://doi.org/10.5281/zenodo.10651900. Other Supplementary information that support this work are available upon request to the corresponding author.

## Code availability

The Mathematica code used to calculate the non-interaction spectra, Wannier center flow, and Hatree-Fock mean-field spectra is available in Zenodo at https://doi.org/10.5281/zenodo.10651900.

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

## Acknowledgements

We would like to acknowledge Liang Fu, Jainendra Jain, Ribhu Kaul, Binghai Yan, Yunzhe Liu, Lunhui Hu and Jiabin Yu for the helpful discussion. The research done by KJY and CXL was primarily supported by NSF through The Pennsylvania State University Materials Research Science and Engineering Center [DMR-2011839]. CXL and BAB acknowledges the support from the Princeton NSF-MERSEC (Grant No. MERSEC DMR 2011750). CXL also acknowledges NSF Grant No. (DMR-2241327). FS was supported by a fellowship at the Princeton Center for Theoretical Science. YX ackonwledges National Natural Science Foundation of China (General Program No. 12374163). BAB was furthermore supported by Simons Investigator Grant No. 404513, ONR Grant No. N00014-20-1-2303, the Schmidt Fund for Innovative Research, the BSF Israel US Foundation Grant No. 2018226, the Gordon and Betty Moore Foundation through Grant No. GBMF8685 towards the Princeton theory program and Grant No. GBMF11070 towards the EPiQS Initiative, and the Princeton Global Network Fund. BAB acknowledges additional support through the European Research Council (ERC) under the European Union's Horizon 2020 research and innovation program (Grant Agreement No. 101020833). PZT was supported by the National Natural Science Foundation of China (Grants No. 12234011 and 12374053).

## Author contributions

C.-X.L. conceived the original idea and supervised the whole project. K.Y performed the theoretical analysis and the numerical simulations with the help of F.S., Y.X., Z.B., B.A.B. and C.-X.L. Z.X. and Y.F. performed the first-principle calculations with the help of P.T. All authors contributed essentially to the manuscript preparation.

## Competing interests

The authors declare no competing interests.
