## [Peer Review File · Nature Communications]

Topological Minibands and Interaction Driven Quantum Anomalous Hall State in Topological Insulator Based Moiré HeterostructuresReviewer #1 (Remarks to the Author):

The authors have investigated a simple model for a thin-film TI in which the surface states are hybridized (because of the thinness of the film), and a Moire potential is supplied using another twisted van der Waals material. It's important to note that the Moire potential is imprinted on the TI thin film. One may ask if such an imprinted Moire potential will have a scale sufficient to induce interesting effects in the hybridized surface states of the TI, but an example DFT calculation using Sb/Sb₂Te₃ suggests this is possible. The section on Hartree-Fock ground states is also valuable and instructive in assessing the effect of Coulomb interactions and how they can drive the system to an AQHE state.

I think in terms of the science, the result is interesting and worthy of publication in Nature Comm as the main conclusion is that nontrivial topologies and also AQHE can be induced in the hybridized surface states. This is of course a celebrated result in twisted bilayer graphene and is looked for in many other twisted systems; perhaps a scheme as the one proposed by the authors is more realizable for AQHE than other twisted bilayers (except for graphene).

I don't have any specific, detailed comments on the calculations - the authors are seasoned enough they know what they are doing, the model Hamiltonians are simple enough, and the DFT calculations as well as the mapping onto model Hamiltonians are carefully done. My main comment is really a lament that the article is extremely dense, something that is exacerbated by all the in-line equations and mathematical expressions. On top of that, the SM are another 23 pages of rather dense calculations. Because there already is a SM 23 pages long, I ask the authors to consider if they can edit or re-write the narrative to better emphasize and describe the physics in a way that is more accessible to a general audience rather than those of us more steepened in Hamiltonians of topological insulators and semimetals. Because the SM is already 23 pages and contains all the mathematical details for those who want to examine them, it seems to me to be somewhat unnecessary to duplicate some of the math in the main manuscript.

Reviewer #2 (Remarks to the Author):

This work focuses on a novel platform of a thin topological insulator with a two-dimensional material stacked on top that supplies a moire potential. The authors combine a theoretical analysis of a generic, effective, model of the low-energy physics with ab-initio calculations of a specific realization.

A new ingredient in this work, relative to other theoretical works on moire topological insulator surface states, is the thinness of the TI. This enables tunneling between surface states, opening up a gaps that are otherwise prohibited. The moire potential is still crucial for opening up gaps at the edge of the BZ. With these two ingredients, the authors are capable of obtaining isolated moire bands.

The authors obtain isolated moire bands in three regions of parameter space I, II, and III. Regime I requires that the moire potential is felt by the bottom layer of the TI, which seems difficult to achieve robustly at least with the depicted setup of a one-sided moire potential (a two sided moire potential would require alignment between the potentials, which may or may not be achievable experimentally). However, Regime II is achievable with a large, but achievable, displacement field. Regime III is quite small and in an unfavorable corner of parameter space.

In all three of the above regions, the authors find topological moire bands, in the sense of a Z_2 Kane-Mele index. Topological moire bands are particularly interesting platforms due to their high tunability and penchant for hosting exotic electronic phases if their bandwidth can be minimized. A symmetry-based argument is given for regime I, suggesting that such bands are a generic feature of this type of platform.

This symmetry argument for regime I, relying on the layer-flipping inversion, appears somewhat distinct from the numerical results which emphasize the role of C_6 symmetry. At zero displacement field, these are actually equivalent demands since $\phi \rightarrow \phi + 1/3$ can be undone by a translation, bringing the system back to an inversion symmetric state. While this is stated in the supplement, it would be helpful to have in the main text so that readers can understand the redundancy in the choice of ϕ immediately.

For regime II, where the large displacement field breaks inversion, it is unclear to me why C_6 is so crucial for the emergence of isolated minibands - if the authors know why it would be a welcome addition to the main text.

The authors find that for sufficiently small twist angles the bandwidth can be made smaller than the interaction strength, and an instability to a quantum anomalous Hall state ensues.

The authors do not discuss the flattening mechanism significantly, and I think it would be helpful for the reader to understand which effective-model parameters are most suitable for flat Z_2 nontrivial bands. From looking at the plots, and the analytical discussion for regime I, it seems that the flattening mechanism relates to the band minimum at Γ being pushed up by the tunneling " m ", and the band maximum at the M point being pushed down by the scale " $m \Delta/vk_M$ ". Based on the residual dispersion, a larger " m ", or larger " Δ/vk_M " would appear to help. The latter seems more feasible experimentally in future work by judicious material choice, small twist angle, or something like pressure to increase Δ . Discussion of flattening physics, and a plot of bandwidth versus some of these parameters, would help future works identify platforms or promising tuning knobs.

The quantum anomalous Hall state order parameter is understood through a basis that makes use of the mirror symmetry, as I learned in the supplement (I believe Eq. S64 is missing a phase factor). In the main text the associated Pauli matrices are quoted as "for the Kramers' pair of CB1" which was not sufficiently descriptive for me to understand the basis alone. Also, it was unclear to me whether the phase that has a large $\rho_{xy}(k)$, the closest competitor to the quantum anomalous Hall phase, is a renormalized version of the non-interacting band structure, or if it is genuinely different in an important way. Does it break any more symmetries, or have a different Fermi surface structure?

The DFT discussion completes this paper well by providing a justification for the parameters chosen in the rest of the paper. I would have been skeptical of a sufficiently large " m " otherwise. One thing that would be good to comment on, however, is the propensity for this system to relax at the moire scale. This is known to happen at small twist angles < 1 degree in TBG due to the energetic favorability of AB stacking versus AA. The authors do not take this into account, which is reasonable for a first study, however it would be nice to comment on its order of magnitude effect relative to graphene if possible, or at least on which stacking regions are energetically preferred (the latter is likely already contained in the DFT calculation?)

Presentation comments:

1) Figure 2 has a few undefined terms, "NI", "SM", and "QSH." I assume "SM" means semimetal and "QSH" means "quantum spin Hall", though "TI" is perhaps more appropriate since there is no spin conservation, and "quantum spin Hall" is not discussed in the main text.

2) The re-use of Regime I,II,III in a different context in figure III is more confusing than it needs to be. Perhaps use A,B,C instead to distinguish it from the parameter regime?

Reviewer #3 (Remarks to the Author):

In this manuscript, the authors consider the physical situation of a 3D TI (in particular, Sb₂Te₃) along with a monolayer of Sb₂ on top, twisted to supply a moire potential on the thin film topological insulator.

To begin, this work finds, in the single-particle realm, a Z₂-nontrivial moire miniband. This is intriguing since *both* surfaces and the hybridization gap are important to this physics. This gap might popularly be thought to be unwanted, but, as this work highlights, there is still important surface physics at work. With gaps opening at Gamma and M, one might wonder how robust the remaining bands are to disorder (and if, for instance, the proposed material has been synthesized with a direct or indirect gap), and some discussion of this would be nice. To state this clearly, this work adds the ultrathin limit and hybridization of surface states into the mix to unveil new surface state physics and topology. Other works, as this manuscript clearly states, have already considering twisting materials on a topological insulator, but in most cases, hard gaps are not found. This work provides hard gaps through hybridization and surprisingly, topology results.

The other major achievement of this work is in finding an interaction-driven quantum anomalous Hall effect. The presence of Coulomb interaction can stabilize the order parameter $\rho_z(k)$ -- the "mirror-polarized state" which leads to Chern bands and a resulting quantum anomalous Hall effect. While the Z₂ work above seems plausible, the stabilization of this interaction-driven phase appears a bit more suspect. The paper does not fully resolve if this is the ground state in some important respects.

On the other hand, there are some large and small problems which this work faces. One of the major criticisms, which the authors should overcome, is the ground state as predicted by Hartree-Fock. One must cast a wide net of order parameters to find the true ground state which (weak) interactions drive and previous work has found (Ref. [32] of the manuscript) that a magnetic texture is the ground state for a single surface. This work explicitly does not consider magnetic textures as a possibility for the interaction driven ground state. A key question one might ask is if the ultrathin limit and hybridization destabilizes a magnetic texture in favor of a Chern band. A confusion also arises with this quantum anomalous Hall effect: It must spontaneously break time-reversal symmetry, yet it is unclear from the text if bands with opposite Chern number occur at the same minimal energy (e.g., Fig. 3ac, can the lower band also carry $C=-1$? And if so, does it look different as dictated by TRS?).

Another point is in regards to the DFT that has been done. First, are the effects topological surface state effects here? A comparison with the bulk band structure would be instructive along with where states are localized within the unit cell. Second, 20 angstrom may not be enough to avoid effects of images (and hence, surface effects which might be spurious). Furthermore, the atoms in the green region of Fig. S15a are frozen giving a rather large amount of strain induced induced on one QL; at the very least, I would worry about claiming equivalence of surface states from the parameter matching which must take place below. It is not clear that this particular setup will, in the low-energy mimic the toy model under consideration. A final point in this regard is the parameter alpha; while it would be nice for the paper to list the numerical parameter fit for all relevant parameters, I would be interested in seeing what alpha turns out to be since I suspect it is quite small or zero -- the bottom layer is quite far away from the Sb atoms on the surface.

To compare this with other work, the thin-film setup is something people in the community would appreciate, but I hesitate that it adds enough past the concept of twistrionics and topological insulators for a broader community to appreciate fully. The work is notable, despite the reservations listed above with both the Hartree-Fock and DFT parameter matching.

A few minor points:

- Important points in the writing are sometimes buried and difficult to find.
- The wall of math entirely inline in Sec. IIB is extremely difficult to read; consider breaking it up with some full-line equations.
- The coloring of blue and orange in Figs. 3ac and Figs. 4cde should be explained in the caption.
- It would be good if it were made more clear in the main text which symmetries are being broken by the Hartree-Fock. Similarly, it would be good to discuss (in the context of the above) why these are the order parameters the authors have chosen (is there a good physical reason?).

Reviewer #1 (Remarks to the Author):

The authors have investigated a simple model for a thin-film TI in which the surface states are hybridized (because of the thinness of the film), and a Moire potential is supplied using another twisted van der Waals material. It's important to note that the Moire potential is imprinted on the TI thin film. One may ask if such an imprinted Moire potential will have a scale sufficient to induce interesting effects in the hybridized surface states of the TI, but an example DFT calculation using Sb/Sb₂Te₃ suggests this is possible. The section on Hartree-Fock ground states is also valuable and instructive in assessing the effect of Coulomb interactions and how they can drive the system to an AQHE state.

I think in terms of the science, the result is interesting and worthy of publication in Nature Comm as the main conclusion is that nontrivial topologies and also AQHE can be induced in the hybridized surface states. This is of course a celebrated result in twisted bilayer graphene and is looked for in many other twisted systems; perhaps a scheme as the one proposed by the authors is more realizable for AQHE than other twisted bilayers (except for graphene).

Reply: Thank you very much for your recommendation for publication of our manuscript in Nature Communications.

I don't have any specific, detailed comments on the calculations - the authors are seasoned enough they know what they are doing, the model Hamiltonians are simple enough, and the DFT calculations as well as the mapping onto model Hamiltonians are carefully done. My main comment is really a lament that the article is extremely dense, something that is exacerbated by all the in-line equations and mathematical expressions. On top of that, the SM are another 23 pages of rather dense calculations. Because there already is a SM 23 pages long, I ask the authors to consider if they can edit or re-write the narrative to better emphasize and describe the physics in a way that is more accessible to a general audience rather than those of us more steepened in Hamiltonians of topological insulators and semimetals. Because the SM is already 23 pages and contains all the mathematical details for those who want to examine them, it seems to me to be somewhat unnecessary to duplicate some of the math in the main manuscript.

Reply: We thank the referee for the suggestion to improve the readability of the manuscript. We follow the referee's suggestions to simplify some discussions about the relationship of the parities at high-symmetry momenta between the conduction and valence minibands and the chiral symmetry in the main text and move these mathematical details to the Supplementary Materials (SM).

Reviewer #2 (Remarks to the Author):

This work focuses on a novel platform of a thin topological insulator with a two-dimensional material stacked on top that supplies a moiré potential. The authors combine a theoretical analysis of a generic, effective, model of low-energy physics with ab-initio calculations of a specific realization.

A new ingredient in this work, relative to other theoretical works on moiré topological insulator surface states, is the thinness of the TI. This enables tunneling between surface states, opening up a gap that is otherwise prohibited. The moiré potential is still crucial for opening up gaps at the edge of the BZ. With these two ingredients, the authors are capable of obtaining isolated moiré

bands.

The authors obtain isolated moiré bands in three regions of parameter space I, II, and III. Regime I requires that the moiré potential is felt by the bottom layer of the TI, which seems difficult to achieve robustly at least with the depicted setup of a one-sided moiré potential (a two sided moiré potential would require alignment between the potentials, which may or may not be achievable experimentally). However, Regime II is achievable with a large, but achievable, displacement field. Regime III is quite small and in an unfavorable corner of parameter space.

In all three of the above regions, the authors find topological moiré bands, in the sense of a Z_2 Kane-Mele index. Topological moiré bands are particularly interesting platforms due to their high tunability and penchant for hosting exotic electronic phases if their bandwidth can be minimized. A symmetry-based argument is given for regime I, suggesting that such bands are a generic feature of this type of platform.

Reply: Thank you very much for carefully reading, appreciating the novelty, and comments in details about our manuscript.

This symmetry argument for regime I, relying on the layer-flipping inversion, appears somewhat distinct from the numerical results which emphasize the role of C_6 symmetry. At zero displacement field, these are actually equivalent demands since $\phi \rightarrow \phi + 1/3$ can be undone by a translation, bringing the system back to an inversion symmetric state. While this is stated in the supplement, it would be helpful to have in the main text so that readers can understand the redundancy in the choice of ϕ immediately.

Reply: We thank the referee for raising this point for clarification. As proved in SM Sec. I B Eq. S35, the moiré potential with certain value of ϕ is equivalent to that with $\phi + \frac{1}{3}$ up to a translation with the vector $a_1^M/3 + 2a_2^M/3$ where a_1^M, a_2^M are primitive moiré lattice vectors. This equivalence only relies on the C_{3v} group symmetry that we impose for the moiré potential. We add a sentence as point 2 in the manuscript change summary to clarify this point.

At zero displacement field, our system has z-directional mirror symmetry, M_z . Consequently, at $\phi=0$ (or equivalent $\phi=1/3$ and $2/3$), the C_6 symmetry leads to the existence of the inversion symmetry I in our system through $I = C_2M_z = C_6^3M_z$. We hope this discussion can clarify the confusion for the reviewer. And we add a sentence in Sec. II B of the main text to emphasize it.

For regime II, where the large displacement field breaks inversion, it is unclear to me why C_6 is so crucial for the emergence of isolated minibands - if the authors know why it would be a welcome addition to the main text.

Reply: We thank the referee for raising this important issue. We first would like to clarify that the emergence of isolated mini-bands does not require C_6 symmetry (See normal insulator phases in Fig. 2a and b in the main text), however, if we hope the isolated mini-bands to be topological, C_6 symmetry is crucial. This aspect can be understood from topological quantum chemistry. When there is C_6 symmetry and the space group is $G P6mm$, the moiré potential forms a honeycomb lattice with two equivalent potential minima at the Wyckoff position 2b (corresponding to the Wyckoff position 1b and 1c if we consider the $P3m1$ group in Fig. 1(c)). As indicated by the symmetry eigenvalues at high-symmetry momenta, the combined lowest two conduction mini-bands CB1 and CB2 correspond to the elementary band representation (EBR) $\bar{E}_1^{2b} \uparrow G$, which is equivalent to putting two s-wave atomic orbitals \bar{E}_1 at the Wyckoff position 2b of the honeycomb

lattice with the space group G $P6mm$. The Wyckoff position 2b of space group is equivalent to the C_6 symmetry related Wyckoff position 1b and 1c labelled with space group $P3m1$ in Fig. 1(c). Once CB1 is isolated from CB2, CB1 itself does not have an atomic limit preserving C_6 symmetry and thus must be topological.[1-2] The breaking of inversion by displacement field adds Rashba spin orbit coupling into CB1 but does not change the above scenario regarding the EBR. If C_6 symmetry is relaxed, CB1 and CB2 instead correspond to the reducible band representation $^1\bar{E}^2\bar{E} + \bar{E}_1 \uparrow G'$ with G' as $P3m1$, correspondingly, they can be gapped without inducing topology. See SM Sec. I C for more details on atomic orbitals. To make this point clear, we explain this mechanism in Sec. II B of the main text.

The authors find that for sufficiently small twist angles the bandwidth can be made smaller than the interaction strength, and an instability to a quantum anomalous Hall state ensues.

The authors do not discuss the flattening mechanism significantly, and I think it would be helpful for the reader to understand which effective-model parameters are most suitable for flat Z_2 nontrivial bands. From looking at the plots, and the analytical discussion for regime I, it seems that the flattening mechanism relates to the band minimum at Gamma being pushed up by the tunneling " m ", and the band maximum at the M point being pushed down by the scale " $m \Delta/vk_M$ ". Based on the residual dispersion, a larger " m ", or larger " Δ/vk_M " would appear to help. The latter seems more feasible experimentally in future work by judicious material choice, small twist angle, or something like pressure to increase Δ . Discussion of flattening physics, and a plot of bandwidth versus some of these parameters, would help future works identify platforms or promising tuning knobs.

Reply: We agree with the referee that the interlayer tunnelling " m " and the moiré potential strength " Δ_1 " are two important parameters for flattening the moiré bands from a perturbative perspective. To support this, we plot the band width of the lowest conduction band CB1 versus these two parameters in Fig. R1, in which we choose the energy unit to be $E_0 = v|b_1^M|$ to scale m and Δ_1 , where b_1^M is the moiré reciprocal lattice vector and v is the Fermi velocity. From this figure, one can see that the bandwidth of CB1 decreases as we increase either m or Δ_1 . In addition, a larger moiré unit cell can reduce b_1^M , as well as E_0 , and thus increase both m/E_0 and Δ_1/E_0 , leading to the narrowing of bandwidth of isolated minibands. We add a sentence to emphasize this point in the main text and a paragraph and Fig. R1 to the SM.

Fig. R1: Bandwidth of CB1 versus the surface state tunneling m and moiré potential strength Δ_1 .

The quantum anomalous Hall state order parameter is understood through a basis that makes use of the mirror symmetry, as I learned in the supplement (I believe Eq. S64 is missing a phase factor). In the main text the associated Pauli matrices are quoted as "for the Kramers' pair of CB1" which was not sufficiently descriptive for me to understand the basis alone. Also, it was unclear to me whether the phase that has a large $\rho_{xy}(k)$, the closest competitor to the quantum anomalous Hall phase, is a renormalized version of the non-interacting band structure, or if it is genuinely different in an important way. Does it break any more symmetries, or have a different Fermi surface structure?

Reply: The reviewer mainly has two questions here: (1) what are the basis wavefunctions of the CB1 mini-bands for the Pauli matrices used in the Hartree-Fock calculations? (2) What's the band structures of $\rho_{xy}(k)$ phase in the Hartree-Fock approximation and how is it different from the non-interacting band structure?

For the first question, we confirm that the Pauli matrices (sigma matrices) in the main text for the order parameter, as the reviewer understood, is written with the horizontal mirror symmetry eigen basis. To clarify that, we add a sentence in Sec. II C of the main text.

For the second question, we clarify that the Hartree-Fock band structure of the $\rho_{xy}(k)$ phase is different from the non-interacting band structure by breaking time reversal symmetry. A comparison of the Hartree-Fock bands for $\rho_{xy}(k)$ phase and the non-interacting bands is shown in SM Fig. S9(a) (also in Fig. R2). Due to the breaking of time reversal symmetry, the Hartree Fock spectrum shows a gap at time reversal invariant momenta Γ and M . It is a linear superposition of two mirror eigen-states (mirror coherent state), and thus breaks a $U(1)$ symmetry of two mirror eigenstates, which is distinguished from the quantum anomalous Hall phase (a mirror polarized state). The symmetry analysis of both phases can be found in SM Table S3. To clarify this point, we modify the discussion about $\rho_{xy}(k)$ in Sec. II C of the main text and add Fig. 3(b)(e) about the Hartree-Fock spectrum for $\rho_{xy}(k)$ in the main text.

For the issue of the phase factor in Eq. S64 in SM $C_{2z}T|u_{\pm i}^{CB1}\rangle = |u_{\mp i}^{CB1}\rangle$, one can indeed add a general phase factor $e^{i\phi}$ in front of $|u_{\mp i}^{CB1}\rangle$. Here we take the convention of choosing the phase $\phi = 0$, and this gauge fixing not influence any essential physics. We modify the discussion in SM Sec. II C to clarify this point.

Fig. R2 (a) The Hartree-Fock spectra (black) for the order parameter $\rho_{xy}(k)$ for the case with $\phi = 0, \alpha = 1, V_0/E_0 = 0$. The blue lines are single particle spectra. (b) The Hartree-Fock spectra (black) for the order parameter $\rho_{xy}(k)$ for the case with $\phi = 0, \alpha = 0, V_0/E_0 = 1.2$. The blue lines are single particle spectra.

The DFT discussion completes this paper well by providing a justification for the parameters chosen in the rest of the paper. I would have been skeptical of a sufficiently large "m" otherwise. One thing that would be good to comment on, however, is the propensity for this system to relax at the moire scale. This is known to happen at small twist angles <1 degree in TBG due to the energetic favorability of AB stacking versus AA. The authors do not take this into account, which is reasonable for a first study, however it would be nice to comment on its order of magnitude effect relative to graphene if possible, or at least on which stacking regions are energetically preferred (the latter is likely already contained in the DFT calculation?)

Reply: We appreciate the reviewer to raise this interesting issue. We below argue that the $Sb_2/2QL$ Sb_2Te_3 heterostructure is easier to relax at the moiré scale with the small twisted angle compared to the twisted bilayer graphene (TBG).

As reported in the previous work [3], evident lattice reconstruction will occur in the TBG if the twist angle is smaller than 0.5° , where the AA stacking regimes decrease while the AB/BA stacking order areas increase. Such a phenomenon results from the competition between the energy gain from stacking configuration transitions and the elastic energy cost induced by the lattice deformation [4]. For a twisted system, if more energy gain is obtained during the transition from a metastable stacking to the most stable stacking, such twisted system will prefer to relax at the moiré scale with a small twist angle. Furthermore, to be similar, if less energy cost is expended during the lattice deformation process, the target twisted system will also prefer to relax at the moiré scale. Therefore, we could evaluate the possibility of the relaxation for the twisted $Sb_2/2QL$ Sb_2Te_3 system in moiré scale from two aspects. One is the energy gain from the transition among various stacking configurations, while the other is the elastic energy cost during the lattice deformation.

The generalized stacking-fault energy (GSFE) can be used to evaluate the energy gain in the stacking configuration transitions, which is defined as the difference of energy density between metastable stacking structures and the most stable stacking structure [5, 6]. Following the method

reported in Ref. 6, we firstly build $\sqrt{3} \times 1$ supercells (dashed rectangle in Fig. R3c) for both bilayer graphene and $\text{Sb}_2/2\text{QL Sb}_2\text{Te}_3$ and choose the AB stacking configuration as the initial stacking order. Then we move the top graphene layer (blue lattice in Fig. R3d) and the Sb_2 monolayer (grey lattice in Fig. R3b) along the x direction (see Fig. R3 b&d) to construct a series of stacking configurations and calculate the GSFE accordingly. Calculated results are shown in Fig. R3a. In the left panels of Fig. R3 b&d, we plot three high-symmetric stacking configurations whose GSFE correspond to the extreme points in Fig. R3a. The maximum of the GSFE of the $\text{Sb}_2/2\text{QL Sb}_2\text{Te}_3$ system is around 172 mJ m^{-2} , which is almost 7 times larger than that of the bilayer graphene system (26 mJ m^{-2}). Therefore, based on DFT calculations, we conclude that the energy gain from the transition among various stacking orders in the $\text{Sb}_2/2\text{QL Sb}_2\text{Te}_3$ system will be much larger than that in the bilayer graphene.

For elastic deformation in the van der Waals stacking structures, the elastic energy (U) can be defined as:

$$U = \sum_{l=t/b} \left[\frac{1}{2} \cdot \frac{E_l v_l}{(1+v_l)(1-v_l)} (\sigma_{ii}^{(l)})^2 + \frac{E_l}{2(1+v_l)} (\sigma_{ij}^{(l)})^2 \right]$$

Here, index (l) refers to the top layer (t) or bottom layer (b); E_l and v_l are Young's moduli and Poisson ratios for each layer. Moreover, $\sigma^{(l)}$ refers to strain tensors for each layer [4]. It is known that the graphene has the largest in-plane Young's modulus. Therefore, under the same strain field, the elastic deformation energy of the bilayer graphene system would be much larger than that of the $\text{Sb}_2/2\text{QL Sb}_2\text{Te}_3$ heterostructure. This result indicates that in-plane lattice deformation is easier in $\text{Sb}_2/2\text{QL Sb}_2\text{Te}_3$ slab model.

To conclude, compared with TBG system, the twisted $\text{Sb}_2/2\text{QL Sb}_2\text{Te}_3$ heterostructure is easier to relax at the moiré scale with the small twist angle due to the larger GSFE and smaller elastic

energy cost during the lattice deformation. We add the discussion to SM Sec. III.

Figure R3. The illustration of the generalized stacking-fault energy for both bilayer graphene (BG) and $\text{Sb}_2/2\text{QL Sb}_2\text{Te}_3$ model. a, The variation of the GSFE of both BG and $\text{Sb}_2/2\text{QL Sb}_2\text{Te}_3$ structures along with the transition of the stacking orders. b, The high-symmetric stacking configurations of the $\text{Sb}_2/2\text{QL Sb}_2\text{Te}_3$ model (left panel) and the side view of the $\text{Sb}_2/2\text{QL Sb}_2\text{Te}_3$ heterostructure (right panel). Only atoms in the light-yellow areas are plotted in the left panel to show every stacking order clearly. c, The illustration of the construction of the supercells (dashed rectangle) from the hexagonal primitive cells (red). d, The high-symmetric stacking configurations of bilayer graphene (left panel) and the side view of the bilayer graphene (right panel).

Presentation comments:

1) Figure 2 has a few undefined terms, "NI", "SM", and "QSH." I assume "SM" means semimetal and "QSH" means "quantum spin Hall", though "TI" is perhaps more appropriate since there is no spin conservation, and "quantum spin Hall" is not discussed in the main text.

Reply: we follow the referee's suggestions and update the manuscript.

2) The re-use of Regime I,II,III in a different context in figure III is more confusing than it needs to be. Perhaps use A,B,C instead to distinguish it from the parameter regime?

Reply: we follow the referee's suggestions and update the manuscript.

Reviewer #3 (Remarks to the Author):

In this manuscript, the authors consider the physical situation of a 3D TI (in particular, Sb_2Te_3) along with a monolayer of Sb_2 on top, twisted to supply a moire potential on the thin film topological insulator.

To begin, this work finds, in the single-particle realm, a Z₂-nontrivial moire miniband. This is intriguing since *both* surfaces and the hybridization gap are important to this physics. This gap might popularly be thought to be unwanted, but, as this work highlights, there is still important surface physics at work. With gaps opening at Gamma and M, one might wonder how robust the remaining bands are to disorder (and if, for instance, the proposed material has been synthesized with a direct or indirect gap), and some discussion of this would be nice. To state this clearly, this work adds the ultrathin limit and hybridization of surface states into the mix to unveil new surface state physics and topology. Other works, as this manuscript clearly states, have already considering twisting materials on a topological insulator, but in most cases, hard gaps are not found. This work provides hard gaps through hybridization and surprisingly, topology results.

Reply: The referee's question is about how robust the gaps of the isolated minibands are when including disorder, which is inevitable in real materials. The topological property of the bands is robust against disorder once the disorder strength is smaller than the band gaps of minibands. Based on our calculations, the mini-band gap is estimated as $0.057E_0 \approx 2.2 \text{ meV}$ for the parameters $\alpha = 1, V_0/E_0 = 0, \phi = 0, |a_1^M| = 28 \text{ nm}$. Therefore, we expect the mini-band topological property is robust against disorder when the disorder strength is smaller than 2.2meV. We add a sentence in the main text to clarify the point.

The other major achievement of this work is in finding an interaction-driven quantum anomalous Hall effect. The presence of Coulomb interaction can stabilize the order parameter $\rho_z(k)$ -- the "mirror-polarized state" which leads to Chern bands and a resulting quantum anomalous Hall

effect. While the Z2 work above seems plausible, the stabilization of this interaction-driven phase appears a bit more suspect. The paper does not fully resolve if this is the ground state in some important respects.

On the other hand, there are some large and small problems which this work faces. One of the major criticisms, which the authors should overcome, is the ground state as predicted by Hartree-Fock. One must cast a wide net of order parameters to find the true ground state which (weak) interactions drive and previous work has found (Ref. [32] of the manuscript) that a magnetic texture is the ground state for a single surface. This work explicitly does not consider magnetic textures as a possibility for the interaction driven ground state. A key question one might ask is if the ultrathin limit and hybridization destabilizes a magnetic texture in favor of a Chern band.

Reply: we thank the referee for raising this important issue. As the referee pointed out, in the previous version of the manuscript, we have only considered the possible uniform order parameters $\rho = \langle C^\dagger(k) C(k) \rangle$, while the non-zero q order parameter $\rho = \langle C^\dagger(k+q) C(k) \rangle$ that breaks the moiré translation symmetry and the time reversal symmetry has not been taken into account.

Below we will discuss the non-zero q order parameter. We would like first to point out two essential differences between the Ref. [32] and our work. The first difference is the different fillings of the minibands. In Ref. [32], the meron magnetic texture is driven by the divergent density of states when the Fermi energy is close to the van-Hove singularities of the non-interacting moiré minibands, while in our work, the lowest energy mini-bands are half-filled and the Fermi energy is away from the van-Hove singularities that correspond to a filling 0.8 of CB1, as shown in Fig. R4. For the second difference, in Ref. [32], only one surface state is considered, while in our work, both the top and bottom surface states are taken into account. Due to spin-momentum locking, the top and bottom surface states have opposite spin helicities. It is found in Ref. [32] that a fixed type of magnetic meron texture can be stabilized by the surface state with one type of spin helicity and is energetically unfavorable for the surface state with opposite spin helicity. In our model, the magnetic meron texture will couple to both surface states with opposite spin helicities, and thus is expected to be less stable as compared to that in Ref. [32].

To further investigate the non-zero q order parameters, we consider the self-consistent calculations of the meron magnetic texture in our model. Here, we consider a magnetic order parameter like Ref. [32] by projecting into eigenstates of non-interaction Hamiltonian

$$M^\alpha(Q) = \langle \hat{S}^\alpha(Q) \rangle = \sum_{k \in \text{MBZ}, n, n'} \langle u_n(k+Q) | S^\alpha | u_{n'}(k) \rangle \langle C_n^\dagger(k+Q) C_{n'}(k) \rangle$$

with $C_n^\dagger(k)$ as the creation operator for eigenstates $|u_n(k)\rangle$ and S^α as the Pauli spin matrices. $\alpha = x, y, z$. Q is the momentum connecting K and K' of the moiré Brillouin zone (MBZ) with von-Hove singularities. The Hartree-Fock mean-field Hamiltonian is

$$H^{HF} = -\frac{1}{2} U^H \sum_{\alpha, Q} \hat{S}^\alpha(Q) M^\alpha(-Q) - E_c[M^\alpha]$$

with $E_c[M^\alpha]$ as the condensation energy. U^H is the Hubbard Coulomb interaction, defined by $U^H = U(|Q|)$ with U as the screened Coulomb interaction in the manuscript. The strength of U is characterized by $U(a_1^M)$ same as Fig. 3(c)(f) in the main text. The details of the self-consistent calculations of the magnetic meron texture can be found in SM Sec. II F.

As shown in Fig. R5, the critical Coulomb interaction strength for nonzero magnetic meron texture is $0.18E_0$, larger than the critical interaction $0.05E_0$ for the QAH phase. In the moiré TI system, the Coulomb interaction is estimated to be $0.13E_0$, which is smaller than that for magnetic meron texture but larger than that of QAH phase. Thus, we conclude that the QAH state, instead of magnetic meron texture, is the ground state at half filling of the lowest conduction band.

In SM Sec. II F, we add a section to discuss our self-consistent calculations of the magnetic

meron texture.

Fig. R4. The spectrum and density of states (DOS) for CB1 (blue lines). The black dotted line is the Fermi energy with Von-Hove singularities, corresponding to the peak of DOS. The red dotted line is the Fermi energy for half filling of CB1.

Fig. R5. The magnetic texture order parameters $|M(Q)|$ versus different Coulomb interaction strength $U(a_1^M)/E_0$ with E_0 as the energy unit for moiré TI.

A confusion also arises with this quantum anomalous Hall effect: It must spontaneously break time-reversal symmetry, yet it is unclear from the text if bands with opposite Chern number occur at the same minimal energy (e.g., Fig. 3ac, can the lower band also carry $C=-1$? And if so, does it look different as dictated by TRS?).

Reply: Yes, the quantum anomalous Hall state indeed spontaneously breaks time-reversal

symmetry. In Fig. 3a of the main text, we only show one QAH ground state with the Chern number $C=+1$, and this state breaks time reversal due to a finite Chern number. In Fig. 3a, the whole Hartree-Fock band structure (the band structure with the Coulomb interaction treated in the mean field way) show the energy splitting of Kramers' pair of mini-bands for CB1 due to the Coulomb interaction. The QAH ground state corresponds to the lower energy bands to be filled while the higher energy band corresponds to the excited spectrum within the mean field approximation. As the lower energy band carries Chern number $C=+1$ in Fig. 3a, this ground state is a QAH state with $C=+1$.

As the single-particle Hamiltonian has time reversal, there should be another QAH state ($C=-1$) with the same Hartree-Fock energy spectrum but the opposite Chern number for each band. As the time reversal is spontaneously broken by Coulomb interaction, these two QAH phases are related by time reversal. We modify the discussion about the QAH ground state in Sec. II C of the main text to clarify the point.

Another point is in regards to the DFT that has been done. First, are the effects topological surface state effects here? A comparison with the bulk band structure would be instructive along with where states are localized within the unit cell.

Reply: We appreciate the reviewer's concern about how topological surface states affect the phenomena reported in this work. In our DFT simulations, the thickness of the Sb_2Te_3 slab is 2 quintuple layers (QL). For such an ultra-thin film, the strong quantum confinement will dominate the electronic structures around the Fermi level, these states around the Fermi level are all contributed by the quantum well states that originate from the strong hybridization of the top and bottom surface states [7].

Furthermore, these quantum-well states will couple with states from Sb_2 monolayer. In Fig. R6, we plot the charge density distribution for the conduction band (CB) edge at the $\tilde{\Gamma}$ point for $\text{Sb}_2/2\text{QL-Sb}_2\text{Te}_3$ with different stackings. Compared with AA and BA stackings, the coupling strength between Sb bilayer with $2\text{QL-Sb}_2\text{Te}_3$ in AB stacking is largest, thus there is relatively

large distribution on Sb_2 monolayer for CB edge state at the $\tilde{\Gamma}$ point. For the AA and BA stacking, the CB edge states at the $\tilde{\Gamma}$ point are mainly localized in the 2QL- Sb_2Te_3 . We add one sentence in the main text and this discussion to SM Sec. III.

Figure R6. Illustration of the electronic structures of the three high-symmetric stacking structures. a, Band structures of the high-symmetric stacking structures (AA, AB, and BA). The black arrows mark the bulk states. b, The charge density distribution on the conduction band edge at the $\tilde{\Gamma}$ point (red circles in Fig. R6a) in the real space of the corresponding slab models. The iso-surface is set as $0.0005 \text{ e \AA}^{-3}$.

Second, 20 angstrom may not be enough to avoid effects of images (and hence, surface effects which might be spurious).

Reply: We appreciate the reviewer's concern about whether the applied vacuum layer is large enough. We performed a convergence test of the vacuum layer size to ensure that the vacuum layer is large enough in this work.

We take the AA stacking structure as an example to test the convergence for the vacuum layer thickness. Firstly, we build a series of AA stacking structures with different vacuum layers (from 12 Å to 28 Å). Then we calculate the work function for each slab model with difference vacuum layer thickness. The results are illustrated in Fig. R7. One can find that the work function almost converges once the thickness of the vacuum layer is larger than 16 Å. We confirm that the vacuum layer of 20 Å is thick enough.

Figure R7. The convergence test of the vacuum layer size. a, Illustration of the dependence of the work function on vacuum layer thickness. The blue arrow marks the thickness of the vacuum layer used in this work.

Furthermore, the atoms in the green region of Fig. S15a are frozen giving a rather large amount of strain induced on one QL;

Reply: We appreciate the reviewer's concern. We ensure that such strain could be neglected after comparing the fully relaxed structure with the corresponding slab model used in the previous manuscript.

We take the slab model with AA stacking as an example. Two relaxation strategies are used to

relax the same initial slab model. One is mentioned in Method part of the previous manuscript, the relaxed structure is labeled as AA-I. The other is to fully relax all atoms' coordinates with the fixed volume of the unit cell and the obtained structure is marked as AA-II. Then we calculate the relative displacement of corresponding atoms in both AA-I and AA-II models to represent the influence of the induced strain on the Sb_2Te_3 thin film.

As shown in Table R1, one can find that the maximum value for the difference of atom positions is 0.009 Å, which is small enough to be neglected. We confirm the methods used in the work are robust. We add the discussion to the SM Sec. III.

Table R1. Cartesian coordinates of atoms in relaxed structures (AA-I/II) and the difference of the coordinates of the corresponding atoms

Atom Index	AA-I			AA-II			Δx (Å)	Δy (Å)	Δz (Å)
	x (Å)	y (Å)	z (Å)	x (Å)	y (Å)	z (Å)			
1	2.128	3.691	22.644	2.129	3.691	22.643	0.001	0.000	0.001
2	0.000	2.462	12.430	0.000	2.461	12.430	0.000	0.000	0.000
3	2.132	3.693	5.000	2.132	3.693	5.003	0.000	0.000	0.003
4	2.132	1.231	15.210	2.131	1.230	15.214	0.001	0.000	0.004
5	2.132	1.231	8.715	2.132	1.231	8.714	0.000	0.000	0.000
6	-0.003	2.460	18.931	-0.002	2.461	18.931	0.001	0.000	0.000
7	0.001	0.000	16.939	0.000	0.000	16.940	0.001	0.000	0.001
8	0.000	2.462	6.729	0.000	2.462	6.720	0.000	0.000	0.009
9	2.132	3.692	10.700	2.132	3.693	10.703	0.000	0.000	0.003
10	2.130	1.229	20.919	2.130	1.229	20.917	0.000	0.000	0.002
11	2.187	1.258	28.156	2.187	1.259	28.157	0.000	0.000	0.001
12	0.055	0.028	26.564	0.055	0.028	26.565	0.000	0.000	0.001

at the very least, I would worry about claiming equivalence of surface states from the parameter matching which must take place below. It is not clear that this particular setup will, in the low-energy mimic the toy model under consideration.

Reply: We thank the referee for raising this concern. We show the TI surface state model Eq.(3) of the main text well describes the low energy conduction bands from DFT by comparing the fitted spectrum (orange lines) from the toy model and the DFT bands (blue lines) in Fig. R8 of all stackings considered. We add the comparison in Fig. 5(c) of the main text and Fig. S19 (e) in SM.

Fig. R8. Comparison of the spectra of the TI surface model and DFT of different stackings AA, AB, BA, AAmAA, AAmAB, AAmBA. Orange lines are spectra of the surface model and blue lines are DFT spectra.

A final point in this regard is the parameter alpha; while it would be nice for the paper to list the numerical parameter fit for all relevant parameters, I would be interested in seeing what alpha turns out to be since I suspect it is quite small or zero -- the bottom layer is quite far away from the Sb atoms on the surface.

Reply: The alpha fitted from DFT calculation is estimated to be 0.16 by comparing the low energy band structure under uniform shift between two materials (See Fig.R8 or Fig. 5(c) in the main text and Fig. S19 in SM). As the reviewer suspected, the value is close to zero because of the relatively large distance between the topological insulator bottom surface and the Sb₂ layer. The fitting result is listed in the paragraph below Eq.(4) in the main text of the manuscript. And the parameters fitted for each stacking are listed in Tab.S5 in SM.

To compare this with other work, the thin-film setup is something people in the community would appreciate, but I hesitate that it adds enough past the concept of twistrionics and topological insulators for a broader community to appreciate fully. The work is notable, despite the reservations listed above with both the Hartree-Fock and DFT parameter matching.

A few minor points:

- Important points in the writing are sometimes buried and difficult to find.
- The wall of math entirely inline in Sec. IIB is extremely difficult to read; consider breaking it up with some full-line equations.

Reply: we have followed the referee's suggestions to simplify the discussion.

- The coloring of blue and orange in Figs. 3ac and Figs. 4cde should be explained in the caption.

Reply: we have followed the referee's suggestions to update the caption.

- It would be good if it were made more clear in the main text which symmetries are being broken by the Hartree-Fock. Similarly, it would be good to discuss (in the context of the above) why these are the order parameters the authors have chosen (is there a good physical reason?).

Reply: we have followed the referee's suggestions to add a sentence about the symmetry breaking of these two states in Sec. II C of the main text.

For our self-consistent calculations with the Hartree-Fock approximation, we project the full model into the low energy bands, either the two-band model for both CB1 minibands or the four-band model with both CB1 and CB2 minibands. Then we classify *all* the uniform order parameters based on the $C_{2z}\mathcal{T}$ symmetry and mirror symmetry (if it exists). This symmetry classification of order parameters simplifies the self-consistent calculations. By randomly choosing the initial guess of order parameters, we look for self-consistent solutions. We have examined different initial guesses, which arrive in the same solution for each type of order parameter. In the updated manuscript, we also examined the non-uniform order parameters with magnetic meron configuration, which have a higher energy and require a larger critical interaction strength, compared to the uniform order parameters, in the reply to the early question of Reviewer #3.

References

- [1] Bradlyn, B., Elcoro, L., Cano, J. et al. Topological quantum chemistry. *Nature* 547, 298–305 (2017).
- [2] Cano, Jennifer, et al. "Building blocks of topological quantum chemistry: Elementary band representations." *Physical Review B* 97.3 (2018): 035139.
- [3] Kazmierczak, Nathanael P., et al. "Strain fields in twisted bilayer graphene." *Nature materials* 20.7 (2021): 956-963.
- [4] Enaldiev, V. V., et al. "Stacking domains and dislocation networks in marginally twisted bilayers of transition metal dichalcogenides." *Physical review letters* 124.20 (2020): 206101.
- [5] Vitek, Vaclav. "Intrinsic stacking faults in body-centred cubic crystals." *Philosophical Magazine* 18.154 (1968): 773-786.
- [6] Zhou, Songsong, et al. "van der Waals bilayer energetics: Generalized stacking-fault energy of graphene, boron nitride, and graphene/boron nitride bilayers." *Physical Review B* 92.15 (2015): 155438.
- [7] Liu, Chao-Xing, et al. "Model Hamiltonian for topological insulators." *Physical Review B* 82.4 (2010): 045122.

Reviewer #1 (Remarks to the Author):

The authors have done a commendable job in responding to all the reviewers' questions or comments. They have also added a non-trivial amount of new information in their responses (some of which has been incorporated in the revised manuscript) to demonstrate the validity of their approach and the robustness of the results. I have nothing further to add and recommend the manuscript for publication.

Reviewer #2 (Remarks to the Author):

The authors have addressed my comments in their entirety, and I have no further strong recommendations or important questions.

This is less of a big deal, but I am a bit confused by their choice to discuss the mirror coherent state as breaking a $U(1)$ symmetry. I think I understand what they mean: the relative phase rotation between the mirror eigenstates is not a symmetry of the state. But I don't think it is a symmetry of the system, or the non-interacting band structure, either. I suppose it is a symmetry of the interacting problem in the "flat band limit," where the non-interacting dispersion is neglected.

But the authors helpfully clarified that the mirror coherent state breaks time reversal symmetry, and this addressed my main previous question.

Reviewer #1 (Remarks to the Author):

The authors have done a commendable job in responding to all the reviewers' questions or comments. They have also added a non-trivial amount of new information in their responses (some of which has been incorporated in the revised manuscript) to demonstrate the validity of their approach and the robustness of the results. I have nothing further to add and recommend the manuscript for publication.

Reply: Thank you very much for your recommendation for publication of our manuscript in Nature Communications.

Reviewer #2 (Remarks to the Author):

The authors have addressed my comments in their entirety, and I have no further strong recommendations or important questions.

This is less of a big deal, but I am a bit confused by their choice to discuss the mirror coherent state as breaking a U(1) symmetry. I think I understand what they mean: the relative phase rotation between the mirror eigenstates is not a symmetry of the state. But I don't think it is a symmetry of the system, or the non-interacting band structure, either. I suppose it is a symmetry of the interacting problem in the "flat band limit," where the non-interacting dispersion is neglected.

But the authors helpfully clarified that the mirror coherent state breaks time reversal symmetry, and this addressed my main previous question.

Reply: We thank the referee for raising this point for clarification. The referee's understanding of the U(1) symmetry is correct. It is not the charge U(1) symmetry, but the relative U(1) phase rotation between two mirror eigen-states (or equivalently spin U(1) phase since two spin states have opposite mirror eigen-values). One should note that our discussion of the relative U(1) phase is only limited to the parameter set when the horizontal mirror symmetry \mathcal{M}_z is present for non-interacting Hamiltonian (the whole single-particle Hamiltonian belongs to D6h group in this case). The presence of \mathcal{M}_z symmetry can simplify the discussion of interaction effect and make the underlying physics transparent. Thus, we mainly focus on the case with \mathcal{M}_z symmetry in the main text, while the case that breaks \mathcal{M}_z symmetry due to an external electric field is mainly discussed in the Supplementary materials. To clarify this issue, we revise the relevant discussion in the main text.